# The influence of upper plate advance and erosion on overriding plate deformation in orogen syntaxes

Matthias Nettesheim[1], Todd A. Ehlers[1], David M. Whipp[2], and Alexander Koptev[1]

[1]Department of Geology, University of Tuebingen, Tuebingen, Germany
[2]Department of Geosciences and Geography, University of Helsinki, Helsinki, Finland

**Correspondence:** Todd A. Ehlers (todd.ehlers@uni-tuebingen.de)

**Abstract.** Focused, rapid exhumation of rocks is observed at some orogen syntaxes, but the driving mechanisms remain poorly understood and contested. In this study, we use a fully coupled thermo-mechanical numerical model to investigate the effect of upper plate advance and different erosion scenarios on overriding plate deformation. The subducting slab in the model is curved in 3D, analogous to the indenter geometry observed in seismic studies. We find that the amount of upper plate advance toward the trench dramatically changes the orientation of major shear zones in the upper plate and the location of rock uplift. Shear along the subduction interface facilitates the formation of a basal detachment situated above the indenter, causing localized rock uplift there. We conclude that the change in orientation and dip angle set by the indenter geometry creates a region of localized uplift as long as subduction of the down-going plate is active. Switching from flat (total) erosion to more realistic fluvial erosion using a landscape evolution model leads to variations in rock uplift at the scale of large catchments. In this case, deepest exhumation again occurs above the indenter apex, but tectonic uplift is modulated on even smaller scales by lithostatic pressure from the overburden of the growing orogen. Highest rock uplift can occur when a strong tectonic uplift field spatially coincides with large erosion potential. This implies that both the geometry of the subducting plate and the geomorphic and climatic conditions are important for the creation of focused, rapid exhumation.

## 1 Introduction

The deformation around orogen syntaxes has been the subject of widespread attention over the last years due to the observed high, sustained, and spatially focused exhumation with rates in excess of $5\,\mathrm{mm\,a^{-1}}$ over million-year timescales. Examples of focused exhumation have been documented for the Olympic mountains of the Cascadia subduction zone (e.g. Brandon et al., 1998; Michel et al., 2018; Adams and Ehlers, 2017, 2018), the Himalayan Syntaxes, Namche Barwa (e.g. Burg et al., 1998; Enkelmann et al., 2011; Stewart et al., 2008) and Nanga Parbat (Craw et al., 1994; Crowley et al., 2009), as well as for the Saint Elias Syntaxis (Berger et al., 2008; Enkelmann et al., 2010; Falkowski et al., 2014) in Alaska. Despite nearly two decades of work, the tectonic- and climate-driven erosional mechanisms responsible for patterns and rates of upper plate deformation in these areas remain debated (Bendick and Ehlers, 2014; Lang et al., 2016; Wang et al., 2014; Whipp et al., 2014; Zeitler et al., 2001).

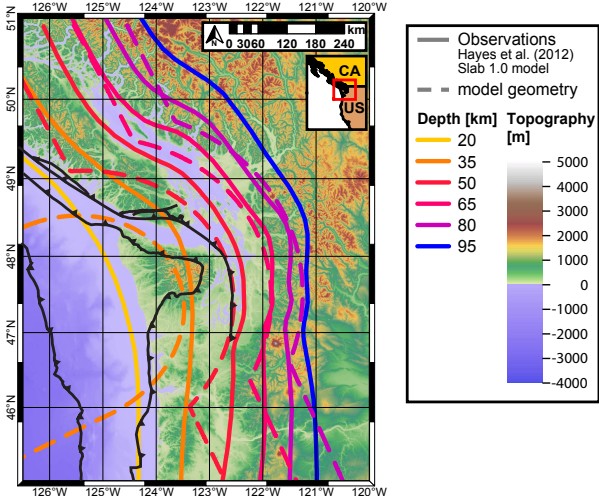

**Figure 1.** Observed subducting plate geometry and simplified model representation for the Cascadia subduction zone. Model geometry was chosen to best conform to the forward bulge of the down-going plate. The rotational shape and straight background slab of the model geometry allow consistent velocity boundary conditions and reduce edge effects. Slab contours from Hayes et al. (2012) Slab 1.0 global subduction zone geometry model, major structures in black from Brandon et al. (1998).

The characteristic 3D indenter geometry of the subducting plate at syntaxes can also be observed, to a lesser extent, at other locations. We use the term "plate corners" in this study to refer to all short, convex bends that separate the longer, straight to slightly concave plate boundary segments. This alternation between bends in the subducting plate and straight segments is a direct consequence of the slab bending that is required to accommodate subduction on a sphere (Frank, 1968; Mahadevan

et al., 2010). Slab curvature and dip angles naturally vary widely across subduction settings observed around the world (Hayes et al., 2012), but common to all of them is their specific 3D geometry. The change in orientation of neighboring subducting slab segments effects flexural stiffening of the connecting region and thus creates a convex forward bend in the subducting slab (Mahadevan et al., 2010), referred to here as an indenter. Figure 1 shows the subducting slab geometry at the Cascadia subduction zone, which served as template to the plate geometry used in this study, although we note that the results presented

here can provide insight into exhumation patterns of other regions where similar plate geometries exist.

In previous studies, numerical modeling has played a central role towards understanding focused exhumation in orogen syntaxes. Work by Koons et al. (2002, 2013) simulated focused exhumation in syntaxes as a function of focused (climate driven) denudation. In their approach, they approximate the subsurface to be of homogeneous composition and define a straight boundary between the subducting and the overriding plate, thus not accounting for the 3D geometry of subducting plates observed

in many syntaxes. Although the link between erosion and uplift through isostasy is well established (Molnar and England, 1990; Montgomery, 1994; Simpson, 2004), their hypothesis of additional positive feedback by thermal weakening of the crust, causing accelerated deformation beneath deeply incised valleys (see also Zeitler et al., 2001), is controversial. Following this,

work by Bendick and Ehlers (2014) considered the effect of the 3D subducting plate geometry, but with simplified upper plate rheology and erosion. However, it has been identified that rheological stratification of the lithosphere (Ranalli and Murphy, 1987; Burov, 2011) is one of the key factors that determines deformation and rock exhumation in convergent orogens, by means of both numerical (e.g. Erdős et al., 2014; Vogt et al., 2017) and analog modeling (e.g. Willingshofer et al., 2013).

Depending on the direction of mantle flow, the amount of slab pull, and whether the subducting slab is anchored in the deep mantle, convergence of plates can be accommodated by both by subduction of the donw-going plate and by advance of the overriding plate towards the trench (e.g. Heuret and Lallemand, 2005; Faccenna et al., 2013). While subduction of the down-going plate commonly accounts for the larger part of shortening (e.g. Sumatra and Java (Chamot-Rooke and Le Pichon, 1999), or Aleutian (Gripp and Gordon, 2002) subduction zones), an additional component of shortening can be taken up by trench or upper plate advance, i.e. migration of the overriding upper plate toward the down-going plate (Heuret and Lallemand, 2005). The Nazca–South America subduction zone (Russo and Silver, 1994; Schellart et al., 2007), and the early stages of the India–Eurasia collision (Capitanio et al., 2010), provide examples of shortening with large amounts of upper plate advance. While Koons et al. (2002, 2013) used a stationary upper plate in their studies, Bendick and Ehlers (2014) considered solely the case of an advancing upper plate.

In this study, we complement previous work by Bendick and Ehlers (2014) and investigate how a rheologically realistic upper plate and slab advance and erosion impact the pattern and rates of exhumation in a generalized plate corner setting. We do this for regions featuring a subducting plate that is bent in 3D and flexurally stiffened at the plate corner. Our first aim is the characterization of upper-plate deformation in convergent settings with a subducting 3D indenter geometry. Since Bendick and Ehlers (2014) employed an advancing upper plate, we additionally focus on possible effects caused by lower plate subduction and the degree to which the upper plate is advancing towards the subduction zone. Our second aim is to understand the effect of erosional efficiency on upper-plate deformation and exhumation by contrasting total erosion (i.e. a constant flat surface) with more realistic fluvial erosion calculated with a surface processes model. It is important to note that we do not aim for exact representation of a specific region, which would require a more detailed and site-specific adjustment of material and thermal properties as well model geometry and boundary conditions, all of which may affect the style of deformation. Rather, we try to assess the impact of an indenter geometry in generic terms in order to understand the underlying mechanisms.

## 2 Methods

### 2.1 Numerical modeling approach

We conduct geodynamical simulations of plate corner subduction with the program DOUAR (Braun et al., 2008; Thieulot et al., 2008), a fully coupled three-dimensional thermomechanical numerical modeling program designed to solve visco-plastic creeping flow equations at the lithospheric scale. Models in DOUAR are defined by a set of velocity boundary conditions, material properties, and model geometry defining the material domains. Bulk velocities are the result of solving the quasi-steady state force balance equations for nearly incompressible fluids with a finite element approach on an octree mesh. Pressure values are derived indirectly from the velocity solution by the penalty method (e.g. Bathe, 1982) and local smoothing is applied

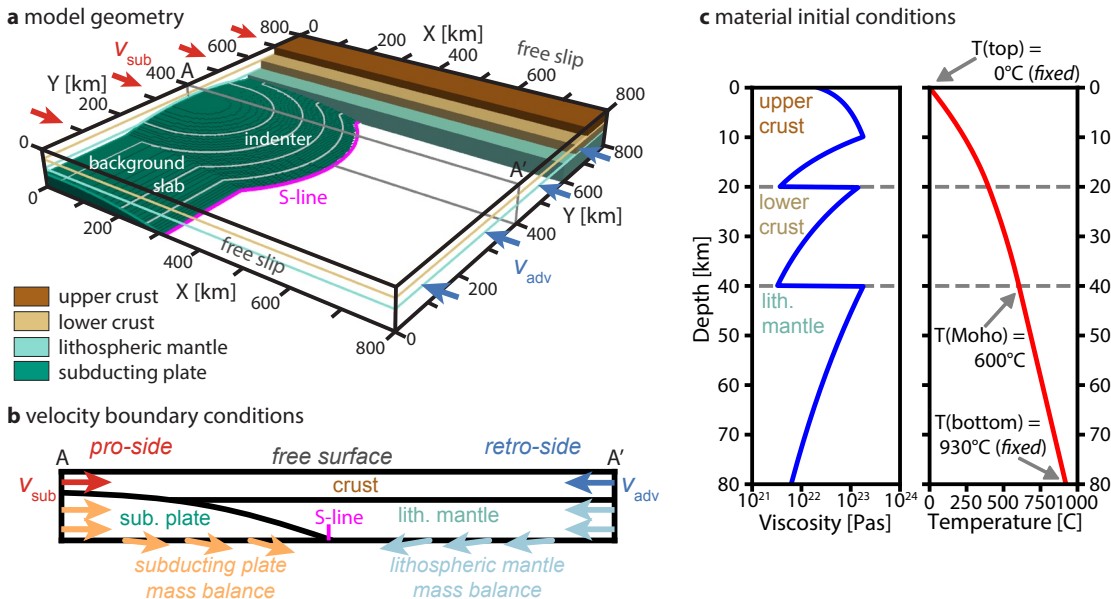

**Figure 2.** Model setup and properties. **a** Cut-out oblique view of the model domain illustrating the geometry and material layers: The overriding plate is divided into an upper and lower crust and lithospheric mantle. The down-going plate is of uniform rigid material. **b** Velocity boundary conditions. Horizontal material influx with $v_{sub}$ from the left ($x = 0\,\text{km}$), and $v_{adv}$ from the right ($x = 800\,\text{km}$), and continued transport towards the S-line along the bottom boundary. Material influx of subducting plate and lithospheric mantle is compensated by rotational motion of the subducting plate and outflux of the lithospheric mantle increasing towards the center, i.e. only crustal influx is added to the domain. Additionally, vertical velocities can be modified by isostasy. Velocity boundary conditions along front and back ($y = 0$ and $y = 800\,\text{km}$) are free slip, top is free surface. **c** Vertical profiles of the overriding plate effective viscosity and temperature initial conditions. Temperature boundary conditions are fixed temperatures at bottom and top, and zero-flux is imposed at all horizontal sides. Detailed material parameters given in table S1.

to avoid small-scale pressure oscillations. Finally, temperature is computed, incorporating the effects of material velocities and thermal properties. Material interfaces and bulk volume properties are stored on a self-adapting cloud of particles advected within the model domain. Additional details of the model and governing equations are given in the appendix.

For this study, we developed a new module that makes use of DOUAR's particle-in-cell approach and permits extraction of the pressure and temperature history (*p-T-t* paths) of particles exhumed at the free surface. Following the methods detailed in Braun (2003), Ehlers (2005), and Whipp et al. (2009), and using the parameters given therein, thermochronometric cooling ages are calculated from these paths to quantify upper plate exhumation patterns and rates over time.

## 2.2 Model setup

### 2.2.1 Geometry

The model domain is $800{\times}800\,\mathrm{km}$ in plan form and $81\,\mathrm{km}$ deep. The element size is $6.25{\times}6.25{\times}1.56\,\mathrm{km}$, corresponding to $128{\times}128{\times}52$ elements for the entire domain. The direction of subduction is parallel to the $x$-axis and the entire setup is symmetric with respect to the $y = 400\,\mathrm{km}$ plane. We refer to the $x = 0\,\mathrm{km}$ and $x = 800\,\mathrm{km}$ model boundaries as *left* and *right*, respectively, and to $y = 0\,\mathrm{km}$ and $y = 800\,\mathrm{km}$ boundaries as model *front* and *back*. The model consists of a layered overriding plate, which is divided into upper and lower crust and lithospheric mantle. We use different geometries for the subducting plate in this study to investigate its effect on overriding plate deformation. The quasi-2D reference geometry is a downward curved straight slab. It is $30\,\mathrm{km}$ high and terminates at $x = 320\,\mathrm{km}$. The standard indenter is modeled after the Cascadia subduction zone (see Figure 1) and is $50\,\mathrm{km}$ high, $350\,\mathrm{km}$ wide and terminates at $x = 400\,\mathrm{km}$. Additionally, we show a narrower and wider indenter (200 and $570\,\mathrm{km}$ wide, respectively).

We simplify the model setup to approximate the effect of a flexurally stiffened indenter by prescribing the geometry of the indenter and subducting plate as fixed through time. Furthermore, we set the shape and motion of the down-going plate to be rotational, i.e. the curved indenter and along-strike continuation of the plate (referred to as the *background slab*) are spheroidal and cylindrical in cross-section, respectively (Fig. 2). This approach minimizes internal deformation and ensures mass balance of the in- and outgoing lower plate.

Additionally, a thin, weak layer on top of the down-going plate is added to ensure partial decoupling, which also reduces indenter deformation (e.g. Willingshofer and Sokoutis, 2009). The thickness of this weak layer is $3.0\,\mathrm{km}$ above the indenter and $4.0\,\mathrm{km}$ above the lower slab in order to compensate for the changing depth to the slab, such that material inflow along the left domain edge is well-nigh horizontal.

### 2.2.2 Material parameters

Corresponding to the focus of this study, we use different material properties for the down-going and the overriding plate. In order to approximate the flexural stiffening of the buckled slab, we use a constant viscosity of $10^{25}\,\mathrm{Pa\,s}$ and no plasticity for the rigid subducting plate. For the overriding plate, we adopt a layered, visco-plastic rheology. The ductile properties of the upper crust, lower crust, and lithospheric mantle are based on wet granite, dry diabase, and olivine viscous flow laws, respectively (Carter and Tsenn, 1987; Hirth and Kohlstedt, 2003; Jadamec and Billen, 2012). The lithospheric mantle viscosity was designed to conform to the known deformation of olivine aggregates (Hirth and Kohlstedt, 2003). Its pressure dependence was eliminated by using parameters from Jadamec and Billen (2012) and recalculation of the remaining parameters (i.e. stress exponent and activation energy) under lithostatic pressure conditions. Brittle parameters are uniform in the crust with $10\,\mathrm{MPa}$ cohesion and an initial friction angle of $15°$. Linear strain softening reduces the friction angle to $5°$ at a strain value of 0.55. In the lithospheric mantle, the friction angle is constant $10°$. An overview of all material parameters is given in table S1.

### 2.2.3 Temperature setup

Model temperatures are set by constant temperature boundary conditions of $0°C$ at the surface and $930°C$ at the bottom. Along all vertical sides, zero flux Neumann boundary conditions are applied. Heat production in the upper and lower crust is chosen to reproduce the thermal structure of Bendick and Ehlers (2014) (see table S1). At startup, temperatures are run to conductive steady state, resulting in a Moho temperature of $600°C$ and heat flux of $25\,\mathrm{mW\,m^{-2}}$ and $70\,\mathrm{mW\,m^{-2}}$ at the model bottom and surface, respectively.

### 2.2.4 Velocity boundary conditions

We model the two end-member cases, where convergence is completely accommodated either by overriding plate advance (*full advance*), or by subduction and accretion (*no advance*), as well as the intermediate case (*half advance*). In order to keep these scenarios comparable, we choose a constant overall convergence rate of $30\,\mathrm{mm\,a^{-1}}$ in all cases (except for model 7 with doubled convergence rate for comparison, see table 1).

These three scenarios translate into different sets of velocity boundary conditions in our models. Subduction and accretion correspond to horizontal influx on the left boundary ($x = 0\,\mathrm{km}$), while overriding plate advance translates to horizontal influx on the right boundary ($x = 800\,\mathrm{km}$) with velocity $v_{\mathrm{sub}}$ and $v_{\mathrm{adv}}$, respectively. Boundary conditions at the front and back ($y = 0$ and $800\,\mathrm{km}$) are free slip. The bottom boundary is set up to ensure the same mass added to the domain in all scenarios. It is separated into two regimes by the S-line (the kinematic boundary between upper and lower plate at the model bottom): to the left, velocities match a downward rotation of the subducting plate, which translates to an increasingly downward direction of motion with velocity $v_{\mathrm{sub}}$ up to the S-line; to the right, horizontal velocity is directed along $x$ with velocity $v_{\mathrm{adv}}$ and an additional linearly increasing material outflux that compensates the influx of material from the right up to the height of the background slab. Finally, material flux through the bottom boundary is also governed by isostasy, which is calculated for an effective elastic thickness of $25\,\mathrm{km}$.

### 2.2.5 Erosion

The top surface is free, with a surface stabilization algorithm based on Kaus et al. (2010) applied. In models 1 through 9, all topography created is immediately removed down to baselevel (*flat* or *total erosion*, see also table 1), which lies $81.0\,\mathrm{km}$ above the model bottom. Under total erosion, rock uplift rate is equal to the exhumation rate (England and Molnar, 1990). The last part of this study contrasts flat erosion with more realistic fluvial and diffusive erosion (models 10–12, table 1). For this, the landscape evolution model *FastScape* (Braun and Willett, 2013) was coupled to DOUAR.

Erosion in FastScape is computed on a regular grid of $0.78\,\mathrm{km}$ resolution with uniform precipitation of $1\,\mathrm{mm\,a^{-1}}$. Erosion constants were $8.0 \cdot 10^{-5}\,\mathrm{m^{-2}}$ for fluvial and $4.0\,\mathrm{m^2\,a^{-1}}$ for hillslope erosion with a stream power exponent of 0.4. The edges at $x = 0\,\mathrm{km}$ and $x = 800\,\mathrm{km}$ are fixed to baselevel and local minima are filled so that each catchment drains to one of those sides.

### 2.2.6 Thermochronometric cooling ages

Thermochronometry determines the time since a mineral has cooled below a characteristic temperature, referred to as the *closure temperature* (Dodson, 1973). This is achieved by the measurement of accumulated decay products in relation to the abundance of radioactive mother nuclides. The loss of decay products – He atoms for the (U-Th)/He method and crystal lattice damage for the fission track method – is thermally activated and specific to each system. At high enough temperatures, all products are lost by diffusion or annealing, respectively. The depth at which temperatures mandate the transition from an open to a closed system is called the partial retention or annealing zone. Above that zone, decay products are accumulated, so their total amount observed at the surface indicates the time since cooling. In order to convert cooling rates into exhumation rates, the geothermal gradient must be known (see e.g. Reiners et al. (2005) for a more in-depth description).

In our numerical models, thermochronometric cooling ages are calculated using tracer particles within the domain, and assuming no deformation for $30\,\mathrm{Myr}$ prior to the model start. Each predicted age represents an integration of deformation from its respective partial annealing or retention zone to the surface (Dodson, 1973; Reiners et al., 2005). Consequently, exhumation of material from the respective zone to the surface is required before the predicted cooling age can be interpreted in a meaningful way. As thermochronometric ages are sensitive to both changes in particle trajectory and thermal gradients, thermochronometric ages will continuously adjust to the evolving geodynamic conditions.

### 2.2.7 Modeling strategy

In this study, we investigate the effects of subduction and frontal accretion ($v_{\mathrm{sub}} > 0$) and upper plate advance ($v_{\mathrm{adv}} > 0$) on upper plate deformation. To study the effect of subduction zone geometry, we first compare straight slab-models with indenter-type models under different velocity boundary conditions. Furthermore, we model different indenter geometries. In the last step, we use a landscape evolution model to study the effect of erosion. We evaluate all models with respect to the resulting rock uplift and strain rates. Additionally, we discuss the models' temporal evolution and the predicted thermochronometric ages at the surface. An overview of all model scenarios is given in table 1.

## 3 Results

### 3.1 Effect of indenter presence and upper plate advance

#### 3.1.1 Overview

Figure 3 illustrates the key features of the indenter geometry effect on upper plate deformation. It shows a snapshot after $4\,\mathrm{Myr}$ simulation time of the indenter-type model under half advance boundary condition (model 5; $v_{\mathrm{sub}} = 50\%$ and $v_{\mathrm{adv}} = 50\%$).

At the model front and back ($y < 200\,\mathrm{km}$ and $y > 600\,\mathrm{km}$), which corresponds to the geometry of the straight slab models (see model 2 in Fig. 4 b4), shortening is accommodated by a lithospheric scale pop-up structure formed by two broad shear zones, indicated by strain rates above $5.0 \cdot 10^{-15}\,\mathrm{s}^{-1}$. These are referred to hereafter as *pro* and *retro shear zone* to the left

**Table 1.** Controlling Parameters Overview

| Model Number | Subducting Plate Geometry | $v_{sub}$ $[\mathrm{mm\,a^{-1}}]$ | $v_{adv}$ $[\mathrm{mm\,a^{-1}}]$ | Upper Plate Advance | Erosion Type | in Figures |
|---|---|---|---|---|---|---|
| 1 | straight slab | 30 | 0 | No | Flat | 4, S1 |
| 2 | straight slab | 15 | 15 | Half | Flat | 3, 4, S1 |
| 3 | straight slab | 0 | 30 | Full | Flat | 4, S1 |
| 4 | indenter | 30 | 0 | No | Flat | 4, 5, 6, S1 |
| 5 | indenter | 15 | 15 | Half | Flat | 3, 4, 5, 6, 7, 9, S1, S2, S3 |
| 6 | indenter | 0 | 30 | Full | Flat | 4, 5, 6, S1 |
| 7 | indenter | 30 | 30 | Half | Flat | S2 |
| 8 | narrow indenter | 15 | 15 | Half | Flat | 7 |
| 9 | wide indenter | 15 | 15 | Half | Flat | 7 |
| 10 | indenter | 30 | 0 | No | Fluvial | S4 |
| 11 | indenter | 15 | 15 | Half | Fluvial | 8, 9 |
| 12 | indenter | 0 | 320 | Full | Fluvial | S4 |

and right of the S-line and labeled ⓟ and ⓡ, respectively. The two shear zones root to the S-line at an angle of ∼60° and comprise shallow dipping detachments and steeply dipping reverse faults. However, due to mostly viscous deformation in the lower crust and lithospheric mantle (see Fig. 2), the shear bands are not as strongly localized there. The particle trajectories on the slices at $y = 50$ and $225\,\mathrm{km}$ show that material is transported horizontally towards the S-line near which paths for crustal material turn upward. In contrast, the lithospheric mantle is gradually subducted. In the central part of the model domain, the active structures change due to the different subducting plate geometry. Above the indenter, deformation is mostly localized in a retro-dipping, upper-crustal detachment (referred to as *indenter detachment*, labeled ①), while the pro and retro shear zones are attenuated. The indenter detachment is better localized (strain rates $>25\cdot10^{-15}\,\mathrm{s^{-1}}$) and is separated from the two previously discussed shear zones, it originates above the subduction interface rather than rooting to the S-line. The material layer boundaries illustrate cumulative deformation by the deviation from the initial horizontal layout.

The differences between the straight trench and indenter models are shown in more detail for all boundary condition scenarios in Figure 4. Since changes in deformation are confined to the indenter's vicinity, this figure depicts strain rates and rock uplift rates along a central slice along dip. Comparing the straight slab models (left-hand panels) shows that deformation and thus rock uplift are always strongest toward the direction of main material inflow. The half advance scenario exhibits a nearly symmetrical pattern of shear zones and rock uplift (Fig. 4 b1 and 4 b2), but in the case of no ($v_{sub} = 100\%$, $v_{adv} = 0\%$, panels Fig. 4 a1 and 4 a2) or full advance ($v_{sub} = 0\%$, $v_{adv} = 100\%$, panels Fig. 4 c1 and 4 c2), deformation shifts to the left or right of the S-line, respectively.

In all models with an indenter geometry, a basal detachment, situated centrally above the indenter, forms the third major structure. It accommodates shortening in the central part of the model and concurrently reduces deformation in the pro and

retro shear zone (Fig. 4 a5–c5). Its position shifts trenchward with increasing upper plate advance. It is strongest directly above the model center and decreases laterally as the height and dip of the subducting plate transition from indenter to the background slab. The straight slab models show a more even and nearly symmetrical upper crustal thinning, whereas material uplift in the indenter models is much stronger and focused at the location of the indenter detachment.

From these three structures, three corresponding zones of rock uplift can be identified. For both the pro and retro shear zones, uplift is localized in the hanging wall of the upper-crustal shallow detachments, forming two continuous bands of uplift in the straight-slab models to the left and right of the S-line, respectively (Fig. 4 a1–c1). The deformation located in the hanging wall of the indenter basal detachment gives rise to the third zone, an elliptical region of rock uplift (Fig. 4 a3–c3).

In all models shown in Figure 4, the mean uplift rate is about $1.3\,\mathrm{mm\,a^{-1}}$, but the distribution of rock uplift strongly depends on velocity boundary conditions. For no slab advance, deformation is strongest on the pro-side, to the left of the S-line. Uplift rates above the indenter are 30% higher than in the straight-slab reference model ($8.3\,\mathrm{mm\,a^{-1}}$ vs. $6.4\,\mathrm{mm\,a^{-1}}$). In the half advance scenario, rock uplift in the straight slab model is evenly distributed between the pro- and retro-side uplift band (3.8 and $4.3\,\mathrm{mm\,a^{-1}}$), while the indenter detachment creates a prominent ellipsis-shaped zone reaching almost twice as fast uplift rates of $7.8\,\mathrm{mm\,a^{-1}}$. Finally, the full advance scenario exhibits lowest peak uplift rates above the indenter of all models ($5.4\,\mathrm{mm\,a^{-1}}$), yet twice as fast as the surrounding area. However, retro-side uplift also reaches equally high rates in both the straight slab and indenter model (5.8 and $5.0\,\mathrm{mm\,a^{-1}}$, respectively). To summarize, all models with indenter geometry exhibit a region of focused and increased rock uplift above the indenter. This effect is small in the full advance model, but much stronger for the no and half upper plate advance scenarios.

Finally, we tested the effect of the chosen convergence rate by running a model with twice the standard value, i.e. at $60\,\mathrm{mm\,a^{-1}}$. It is shown in comparison to the standard half advance model in Figure S2. There is little difference in the relative distribution of strain and rock uplift between the two models, except at the front and back, where uplift between the pro and retro shear zones is increased in the fast convergence model.

### 3.1.2 Temporal evolution under different plate motion scenarios

As simulation time progresses, the deformation patterns and rock uplift change differently for the three scenarios. Figure 5 shows strain rates at $8\,\mathrm{Myr}$ modeling time and the gradual evolution of surface rock uplift rates along the central slice ($y = 200\,\mathrm{km}$). In both the no and half advancing slab scenario, the peak in uplift above the indenter increases and reaches a quasi-steady state. In the case of full slab advance, it uplift decreases after reaching a maximum at $\sim 4\,\mathrm{Myr}$. In this scenario, the strong and lasting increase can be observed in the right uplift band instead. These changes in uplift are also reflected in the position and modes of shear zones (compare Figs. 3 and 4 showing strain rates at $4\,\mathrm{Myr}$ simulation time with strain rates at $8\,\mathrm{Myr}$ in Fig. 5). Without slab advance, the indenter basal detachment slowly shifts to the right over time, while keeping its general shape. In contrast, additional shear zones develop alongside the basal detachment for the half advance scenario. In the case of full slab advance, however, motion across the indenter detachment almost ceases in favor of a reverse fault dipping in the pro-direction.

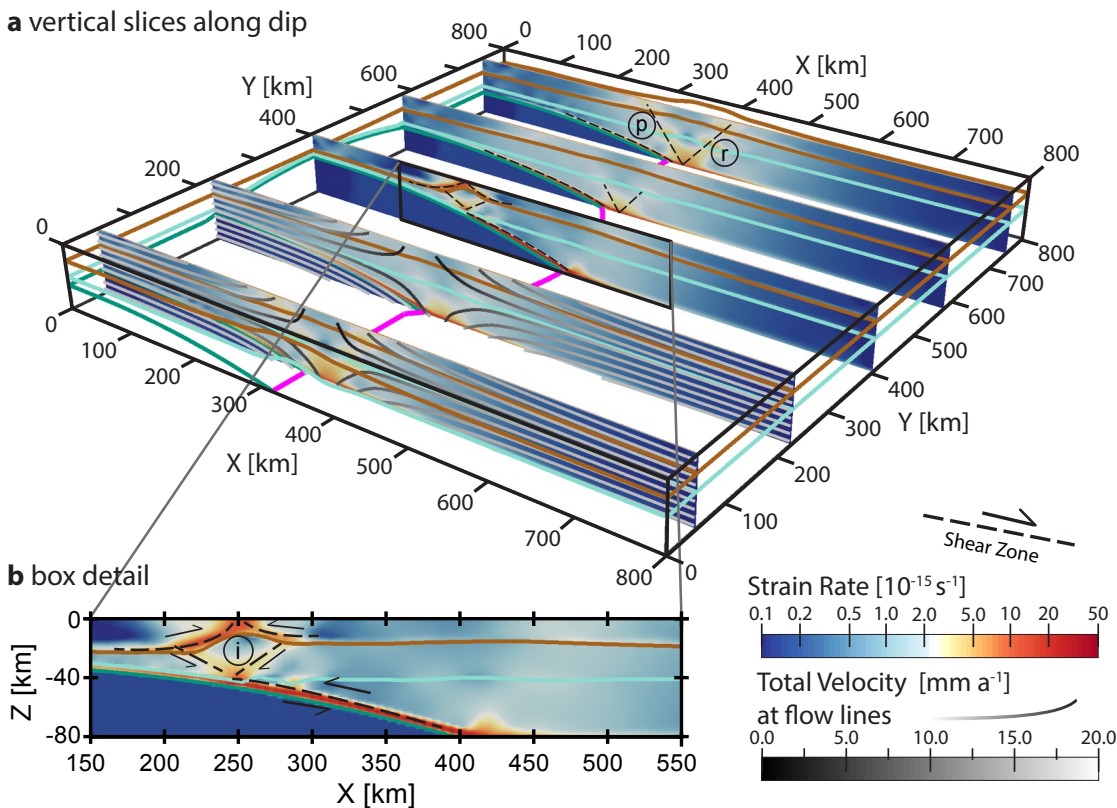

**a** vertical slices along dip

**b** box detail

Strain Rate [10⁻¹⁵ s⁻¹]

Shear Zone

Total Velocity [mm a⁻¹]
at flow lines

**Figure 3.** Vertical slices along dip after 4.0 Myr simulation time showing the second invariant of the strain rate tensor and particle trajectories for the indenter geometry under half advance boundary conditions. Brown, light green and dark green lines denote the material layer boundaries between upper, lower crust and lithospheric mantle, respectively; magenta marks the S-line. Pro and retro shear zone are labeled ⓟ and ⓡ, indenter detachment with ⓘ. Note that results are symmetrical about the $y = 400\,\text{km}$ plane. **a** Slices along dip in 3D: Deformation at the front and back is weakly localized in two symmetrical shear zones rooting to the S-line, forming a lithospheric-scale pop-up structure. Above the indenter, shortening is accommodated by a newly formed basal detachment replacing the steeply dipping shear zones. **b** Detail inset shows the strain rates along the central slice.

### 3.1.3 Prediction of thermochronometric ages

Figure 6 shows a comparison of resulting zircon (U–Th)/He cooling ages and rock uplift rates for models 4–6 (indenter geometry with flat erosion) after 8 Myr of simulation time. In all cases, the distribution of cooling ages matches that of rock uplift rates at the surface (see corresponding plots in Figures 4 and 5). This can also be observed in other thermochronometric systems as well as physical exhumation parameters, which are shown in Figure S3 for reference. In Figure 6, the shift in deformation from pro to retro shear zone with increasing upper plate advance is clearly visible. Additionally, the concentration of young ages above the indenter can be seen in all scenarios. Even in the case of full upper plate advance (Fig. 6 c), where

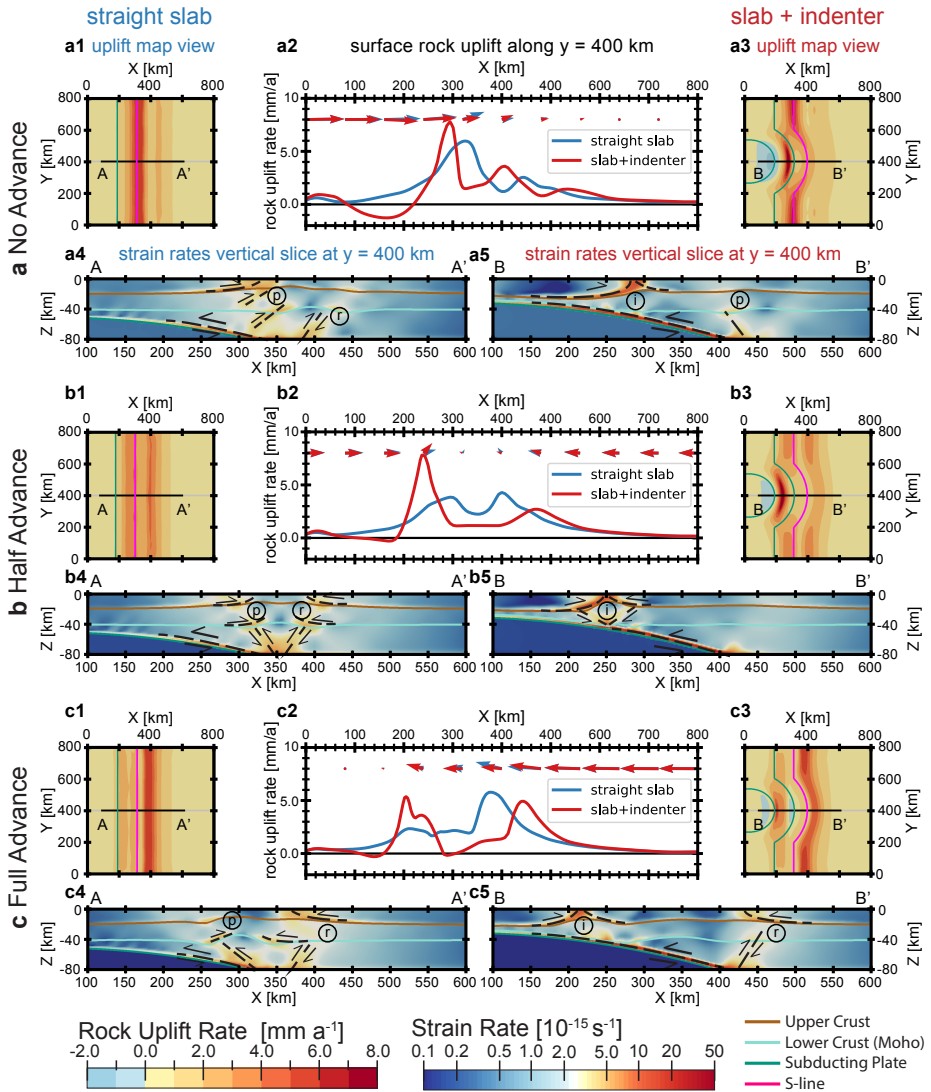

**Figure 4.** Effect of indenter geometry on rock uplift and strain rates for models 1–6 (Table 1). For each of the three upper plate motion scenarios, panels 1 and 3 show the plan view distribution of rock uplift rates (vertical component of velocity) for the straight slab and indenter geometry, respectively. Magenta lines mark the position of the S-line and dark green are indenter contours at 40 and 60 km depth. Panels 2: rock uplift rates at the surface along dip at $y = 400$ km (grey line in panels 1 and 3). Arrows indicate orientation of total velocity. Panels 4 and 5 show strain rates for vertical slice along dip at $y = 400$ km (grey line in panels 1 and 3). Brown, light green and dark green indicate the material boundaries between upper plate, lower plate, and lithospheric mantle, respectively. Pro and retro shear zone are labeled ⓟ and ⓡ, indenter detachment ⓘ. In all scenarios, the indenter gives rise to an ellipsoidal region of increased uplift above its apex, caused by the indenter detachment that accommodates shortening in the indenter's vicinity.

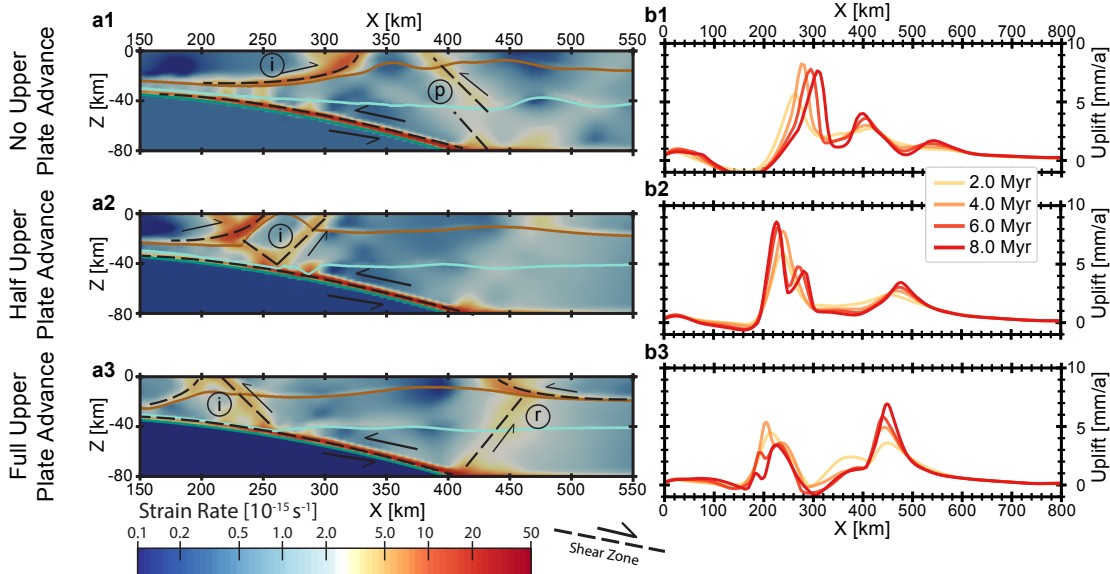

**Figure 5.** Temporal evolution of the no, half, and full upper plate advance indenter geometry models (4–6). **a** Strain rates on the $y = 400\,\text{km}$ cross-section at $8\,\text{Myr}$ modeling time (cf. Fig. 4 showing same slices at $4\,\text{Myr}$). **b** Time slices of surface rock uplift rates along $y = 400\,\text{km}$. No upper plate advance (model 3; top panels) scenario shows an indenter detachment in dynamic steady state, which slowly migrates to the right. In the half advance scenario (model 4; central panels), uplift above the indenter likewise saturates, but its position is stable and additional shear zones form in the last stage. Lastly, the full advance case (model 6; bottom panels) exhibits most uplift at the right uplift band, which continuously increases, whereas the indenter uplift ceases again after reaching a peak at $4\,\text{Myr}$ simulation time.

most deformation is accommodated by the shallow detachment of the retro-shear zone, this decrease in ages towards the area above the indenter is visible in the zircon (U–Th)/He ages.

## 3.2 Effect of indenter geometry

In addition to comparing a straight slab to an indenter-type subducting slab, we investigate the indenter's effect on upper plate deformation by modeling different geometries. Figure 7 shows the distribution of rock uplift rates for both a narrower and wider indenter geometry compared to the standard/reference geometry discussed previously. For all models, the basic effect of the indenter is the same in that an ellipsoidal region of increased rock uplift is generated above its apex around $x \approx 225\,\text{km}$, $y \approx 400\,\text{km}$. The maximum uplift rates, however, increase with indenter width. While the narrow indenter reaches only $6\,\text{mm}\,\text{a}^{-1}$, rock uplift rates increase to $8\,\text{mm}\,\text{a}^{-1}$ and even $9\,\text{mm}\,\text{a}^{-1}$ for the standard and wider indenter, respectively. Furthermore, in the narrow indenter model, the region of high uplift is not as clearly separated from the uplift caused by the pro shear zone.

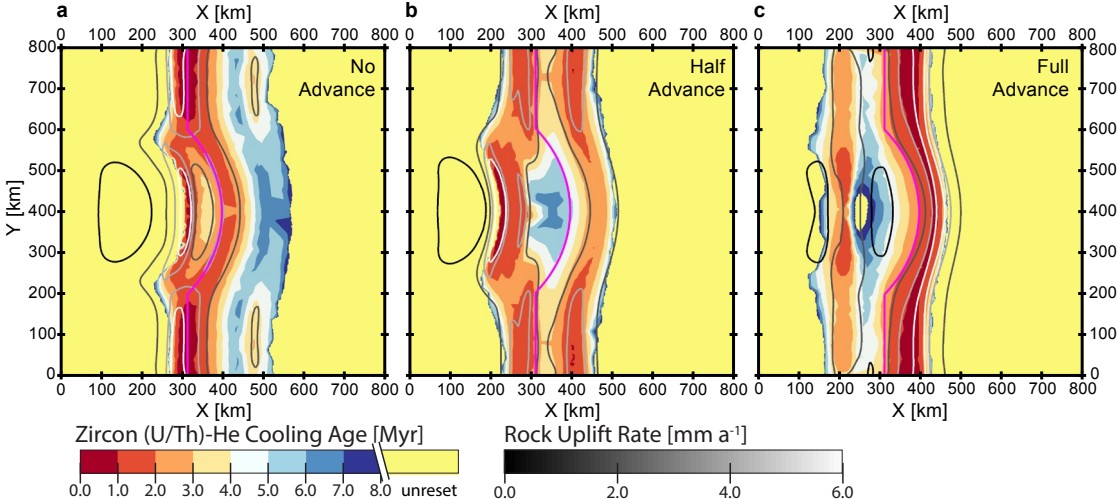

**Figure 6.** Comparison of zircon-(U–Th)/He cooling age for the three standard indenter flat erosion scenarios (models 4–6). In all cases, cooling ages match well with the distribution of rock uplift (see figures 4 and 5). The shift in deformation focus toward the direction of main material influx is clearly visible by the concentration of young ages shifting from the left to the right of the S-line with increasing upper plate advance.

### 3.3 Effect of variable erosion

For the standard indenter geometry, three additional experiments coupled to a landscape evolution model (*FastScape*, Braun and Willett (2013)) were performed under no, half and full upper plate advance boundary conditions. The distribution of strain rates and large-scale particle trajectories, as described in section 3.1.1, are only weakly affected by this switch to fluvial
erosion. However, the creation of topography and the resulting variations in lithostatic pressure cause changes in upper crustal deformation as illustrated by Figure 8, which shows the half advance fluvial erosion model (model 11).

Figure 8 a shows the topography created after $6.0\,\mathrm{Myr}$ simulation time, with highest peaks towards the model edge and lower topography and a curved flank above the indenter. The distribution of rock uplift (Fig. 8 b) forms two broad bands situated on the flanks of the forming orogen. The left band follows the shape of the slab contour at $50\,\mathrm{km}$ depth, with a slight increase in
uplift rate above the indenter apex, the right one is only slightly curved and located at $x = 500\,\mathrm{km}$. In comparison with flat erosion (Fig. 8 d), rock uplift zones are much wider, especially towards the model front and back, and uplift rates are reduced by roughly half. Additionally, there are strong local variations, with peaks in rock uplift situated in river valleys. These local maxima (up to $3\,\mathrm{mm\,a^{-1}}$ compared to an average $1.0\,\mathrm{mm\,a^{-1}}$) correspondingly show deeper exhumation, as can be seen from Figure 8 c. The two uplift peaks in the marked catchments show the deepest exhumation ($\sim 8\,\mathrm{km}$). They are situated above the
indenter and within the region of deepest exhumation in the flat erosion scenario, shown in Figure 8 e for comparison. They also comprise locations where the youngest thermochronometric ages are predicted (Figure 9). The fluvial erosion models for no and full upper plate advance (shown in figure S4) a close correspondence to the flat erosion models, as well. However,

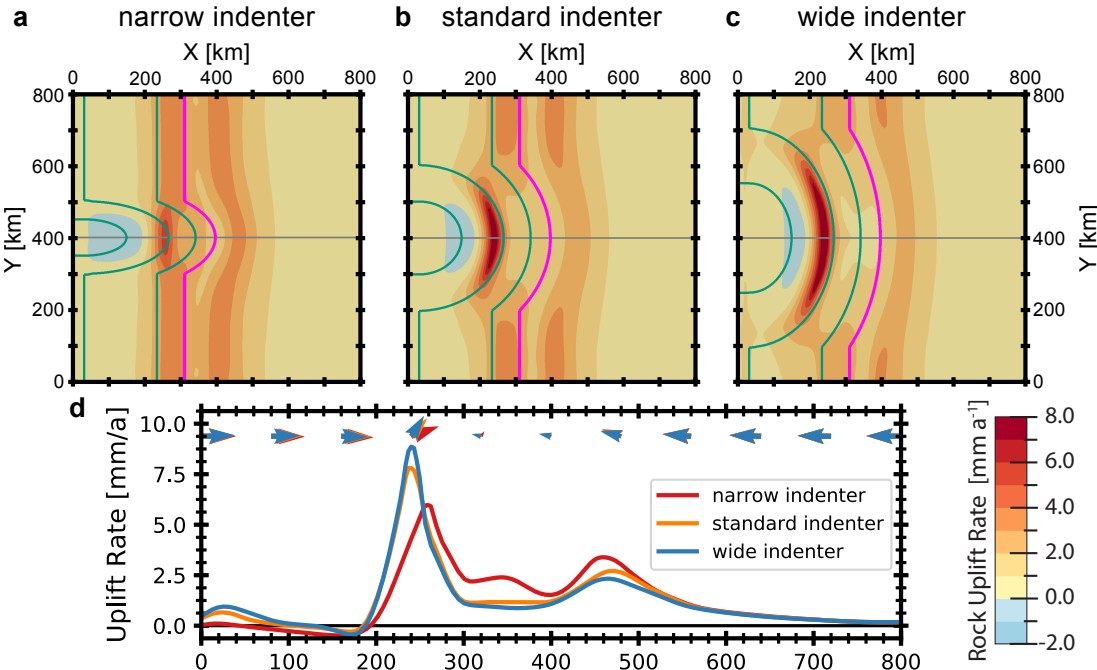

**Figure 7.** Comparison of different indenter geometries under half advance boundary conditions. **a–c** map of surface rock uplift rates for narrow, standard, and wide indenter geometry. **d** rock uplift rates along a central cross section (grey line in panels a–c). Basic distribution of uplift is similar in all models with a region of increased uplift above the indenter apex at $x \approx 225\,\mathrm{km}$, $y \approx 400\,\mathrm{km}$. Maximum rock uplift rates increase with indenter width.

only much smaller catchments cut across regions of increased tectonic uplift. Consequently, there is no local increase in rock uplift as substantial as observed in the half advance scenario. In summary, fluvial erosion leads to variation of rock uplift and exhumation on the catchment scale and maxima in rock uplift roughly $60\,\mathrm{km}$ in diameter, while regions of high uplift in the flat erosion scenarios extend $\sim 250\,\mathrm{km}$ along strike. These zones of deepest and fastest exhumation are situated above the indenter apex.

## 4 Discussion

### 4.1 Summary of model results

The simulations with a rigid subducting plate presented here indicate that shortening is accommodated by a lithospheric-scale pop-up structure formed by two broad shear zones rooting to the S-line (pro and retro shear zone). Each shear zone comprises shallow detachments and steeply dipping faults. The shear zone oriented towards the direction of material influx is expressed more strongly. In experiments with an indenter bulging forward from the subducting plate (models 4–12), a more localized

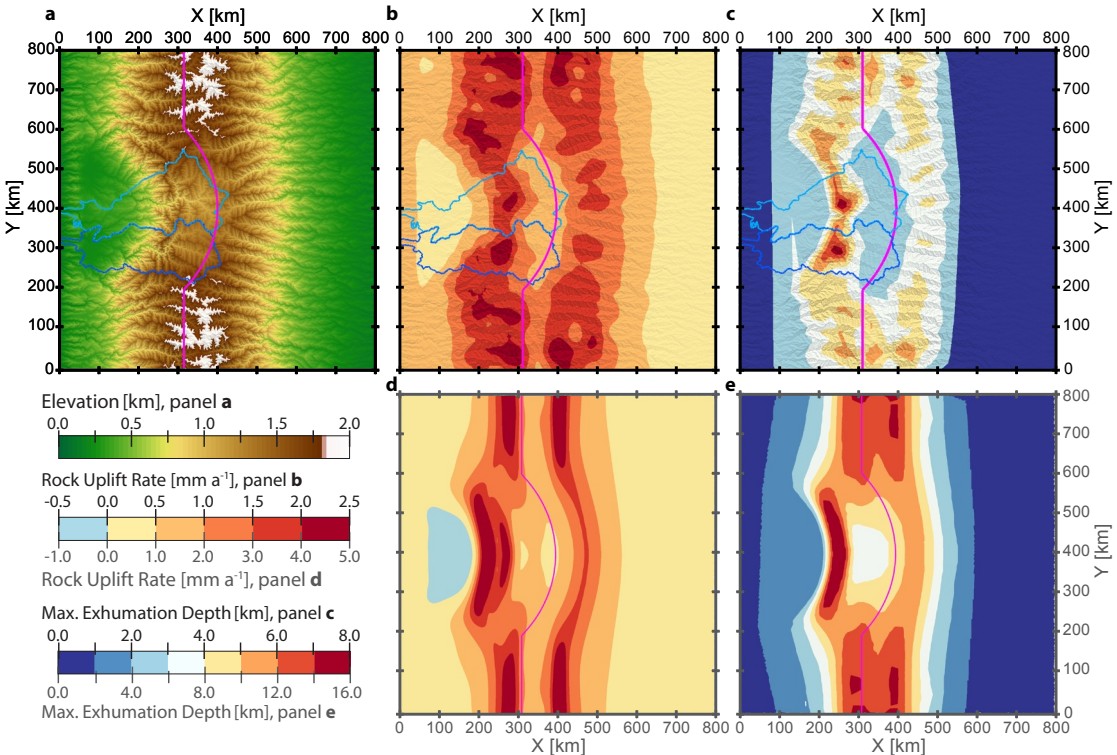

**Figure 8.** Comparison of rock uplift and exhumation depths with fluvial versus total erosion. **a** Surface elevations, **b** rock uplift rates, and **c** exhumation depth for model 11 (half advance fluvial erosion) after $6.0\,\mathrm{Myr}$ simulation time. Uplift is focused along the flanks of the orogen, with the two maxima in exhumation situated in river valleys of high erosion potential. Deepest exhumation occurs in the two uplift foci situated above the indenter ($x = 265\,\mathrm{km}$, $y = 410\,\mathrm{km}$ and $x = 250\,\mathrm{km}$, $y = 300\,\mathrm{km}$), situated within the catchments outlined in blue. Predicted thermochronometric ages are: apatite (U–Th)/He: 1.3 and 1.1 Ma; apatite fission track: 1.9 and 1.65 Ma; and zircon (U–Th)/He 4.2 and 4.1 Ma for the upper (light blue catchment) and lower (dark blue) hotspot respectively. **d** and **e** show the flat erosion results for rock uplift rates and exhumation depth for comparison. Note the range is increased by a factor of 2 for those two plots (gray labels).

basal detachment forms above its apex (indenter detachment) and accommodates shortening there. Concurrently, shear across the pro shear zone is strongly reduced. This effect is limited to the indenter's vicinity, as can be seen from models with a narrower and wider indenter (Fig. 7). These shear zones give rise to three regions of rock uplift in the flat erosion scenarios: the shallow crustal detachments of the pro and retro shear zones create two bands that follow along the trace of the S-line, offset to the left and right, respectively. The third zone of high uplift, created by the indenter detachment, is shifted further trenchward (left) and forms an isolated, elliptical region of uplift that is stronger in the no and half upper plate scenarios (e.g. Figs. 4 and 5).

## 4.2 Model caveats and limitations

Our model setup is generalized to include the first-order features of plate corner settings, but is not designed to reproduce a specific region, given limitations due to numeric resolution and large uncertainties on rheologic and thermal parameters of any particular region. Moreover, rock uplift rates in our model are overestimated for flat erosion simulations (models 1–9, shown in Figs. 3–7), which inhibits the creation of topography that would exert an isostatic counter-force to material uplift. With these caveats in mind, rock uplift rates in the flat erosion models should be seen as upper bound in cases of extremely high erosion. Corresponding to the high rock uplift rates, our predicted cooling ages are very young, but nevertheless consistent with the range of exhumation rates reported at syntaxes: $\sim 1\,\mathrm{mm\,a^{-1}}$ in the Olympic mountains (Brandon et al., 1998; Michel et al., 2018) 3–5 $\mathrm{mm\,a^{-1}}$ in SE Alaska St. Elias range (Enkelmann et al., 2016), and 5–9 $\mathrm{mm\,a^{-1}}$ in the eastern Himalayan syntaxis (Enkelmann et al., 2011; Lang et al., 2016). It is also worth noting that in the fluvial erosion simulations, uniform precipitation was used throughout the model run. While this is justifiable for our study of contrasting erosional efficiency, it has been shown that changes in precipitation by orographic effects strongly influence the distribution of rock uplift and consequently orogen dynamics (Willett, 1999). Despite these limitations, several general lessons emerge from our simulations that have bearing on understanding exhumation processes in plate corner settings. In the following, we will discuss key aspects and mechanisms of exhumation at plate corners.

## 4.3 Effect of indenter presence and geometry

The variation in subducting plate geometry from background slab to the indenter causes different structures to evolve to accommodate shortening (e.g. Figs. 3 and 4). Through this change in structures along strike, localized exhumation is created. Once deformation is localized, both strain and thermal weakening contribute to intensify shear across those structures (see temperatures in figure S1) and focus deformation even further in that region, thereby increasing uplift rates. In our models, the indenter detachment is created by the shallower slab and lower dip angle of the indenter, which exerts a stronger traction on the overriding plate, both forward and downward (see Fig. 4 and velocities in Fig. S1). Additionally, the indenter detachment partially takes up strain that would otherwise be accommodated by the pro shear zone. Strain transfer between these systems occurs along the weak subducting plate interface. Along the background slab, the interface's lower dip angle seemingly makes strain transfer less favorable (cf. Fig. 4). The extent of the indenter detachment is limited to the indenter width, but a minimum size is required to observe its effect. For the narrow indenter, material is displaced laterally rather than exhumed. This is also evident by the higher uplift rates for the wider indenter, where lateral displacement is most strongly impeded. Detachment-like shallow faults dipping towards the subducting plate have been reported for the Andean orogen (e.g. Horton, 2018b). It is important to note that the lateral variation in depth to the subducting slab in our models may contribute to the changing style of deformation. Still, the Cascadia subduction zone, template to our model geometry, is an example of such a change along strike. Above the indenter bulge, the Hurricane ridge fault warps from offshore to on-land and exposes the accretionary wedge in the Olympic mountains, directly above the indenter apex (Figure 1 and 9, Brandon et al. (1998)). This is supported by work

of Calvert et al. (2011), who deduce strongly increased underplating of sedimentary material above the indenter bulge from seismic tomography.

## 4.4 Effect of upper plate advance

Many numerical models of subduction processes use a fixed overriding plate with respect to the trench position (see e.g. Koons et al., 2010; Braun and Yamato, 2010; Willett, 1999). In contrast, Bendick and Ehlers (2014) used a stationary indenter and advancing upper plate. Our models explore how focused uplift created by an indenter geometry varies with respect to different velocity boundary conditions. The indenter detachment is more active in the no and half upper plate advance scenarios, corresponding to the behavior of the pro shear zone. In the full advance scenario, however, the indenter detachment is only of transient nature, and rock uplift rates above this feature reach barely half those seen in the other models (Fig. 5). Since localization of rock uplift by the indenter detachment corresponds to a different accommodation of shortening along strike caused by the indenter geometry, it seems evident that sufficient deformation in the indenters' vicinity is required to make those different structures observable. As much of the overall shortening in the full advance case is taken up by the retro-shear band, even increasing by time through thermal and strain weakening, strain transfer along the weak subduction interface and thus uplift above the indenter is strongly reduced. The increased deformation in the retro shear zone – associated rock uplift rates increase by $\sim$40% from 4 to 8 Myr simulation time – leads to subsidence above the indenter (see Figure 5 a3), because the correspondingly reduced material flow along the subducting plate interface can no longer support the thinned crust, which then relaxes isostatically. This conforms to observations of the Chilean forearc, where compression has induced flexural subsidence by thrust or reverse faulting in the late Cenozoic (see Horton, 2018a, and references therein).

Additional effects of the different velocity boundary conditions can be observed between the no and half upper plate advance scenarios. Both flat erosion models reach a dynamic steady state at $\sim$4 Myr with rock uplift rates of $\sim$8 mm a$^{-1}$ above the indenter. Although maximum uplift rates are about the same, the region of focused uplift above the indenter appears more prominent in the half advance case. This is due to the even distribution of shear between the pro and retro shear zones in the half advance scenario, which creates uplift over a wider area at consequently lower rates. Their respective rock uplift rates at the front and back reach only 4 and 5 mm a$^{-1}$ in the half advance scenario, whereas uplift rates in the no upper plate advance model reach up to 9 mm a$^{-1}$. Finally, the indenter detachment in the no advance scenario is slowly shifting to the right (away from the trench). In the half advance model, this motion is prevented by opposing material inflow, focusing deformation and thus stabilizing deformation through increased thermal weakening. The exhumation of lower crustal material at 8 Myr simulation time (panel b4 in Figure 5) illustrates this focusing effect well.

Note that the overall rate of convergence has only limited effect on the distribution of deformation. Naturally rock uplift is twice as fast for the doubled convergence rate (see Fig. S2), but the relative distribution is mostly the same. Small deviations in strain rate fields stem from different isostatic adjustment and the strain-dependent material viscosity.

As expected, predicted cooling ages and exhumation parameters calculated conform with the modele velocity fields and offer an additional option to illustrate the distribution of deformation in the overriding plate. They provide the best option to relate model results to natural observations. However, rock uplift in our models is exceptionally fast due to flat erosion, which

prevents the creation of topography that would counteract rock uplift. Given this, it is important to take the resulting values as only a general representation of what could be observed in natural systems, and to focus on the age pattern rather than the absolute values.

## 4.5 Sensitivity of exhumation to different erosion parameters

The distribution of rock uplift is quite similar for the flat and fluvial erosion model runs. Nevertheless, the growing topography means an increased lithostatic overburden, causing both a general reduction in rate and redistribution of rock uplift. In general, rock uplift is spread out over a wider area than in the flat erosion models. Additional spatial variations in rock uplift occur on the catchment scale. Strong erosion removes lithostatic overburden and engenders higher uplift rates, mostly so in steep and sufficiently large catchments along the orogen flanks. Under half advance, two rivers with large upstream catchment area cut across the region of localized uplift above the indenter apex (Fig. 8). It is there, where strong tectonic forcing coincides with a high erosion potential such that the deepest and fastest exhumation is observed. Both other fluvial erosion models (no and full advance) show variation in uplift on the catchment scale as well (Fig. S4), but lack comparably large rivers transecting regions of high tectonic uplift. Thus, no equivalent localization effects can be observed in these models. From these results, we hypothesize that both focused deformation and focused erosion are necessary in order to form a region of localized and rapid uplift. The curved 3D-geometry of the subducting plate at plate corners sets the stage, as proposed by Bendick and Ehlers (2014), but only when this concurs with a sufficiently high erosion potential, a spatially limited area of intense uplift can be formed (see Zeitler et al., 2001). Prime examples for this are the Himalayan syntaxes, where the Yarlung and Ganges river, respectively, cut across the Himalayan range and steepen rapidly (Finlayson et al., 2002). In combination with active crustal-scale structures (Burg et al., 1998; Schneider et al., 1999), this has given rise to spatially limited, rapid exhumation as indicated by very young thermochronometric cooling ages (Zeitler et al., 1993; Winslow et al., 1996; Burg et al., 1998; Malloy, 2004; Zeitler et al., 2014).

Figure 9 shows model results in comparison to cooling age observations of the Cascadia region, which served as template for the subducting plate geometry used in this study (see Fig. 1). In general, the modeled distribution of rock uplift rates reflects topographic highs and lows. Moreover, the concentration of young cooling ages (Apatite (U-Th)/He $< 3\,\mathrm{Myr}$) and inferred uplift rates of $\sim 1.0\,\mathrm{mm\,a^{-1}}$ observed in the Olympic mountains (Brandon et al., 1998; Michel et al., 2018), correspond quite well to the peak in rock uplift rates and cooling ages predicted from the fluvial erosion model. The length scale of variations in rock uplift introduced by fluvial erosion also seems to be in agreement with the observed cooling age data, which show young and old ages often in close proximity. Nevertheless, our model overestimate rock uplift rates with large areas above $1\,\mathrm{mm\,a^{-1}}$ that do not agree with Apatite Fission track ages $> 20\,\mathrm{Myr}$ on Vancouver Island, for example. The strong gradient of observed cooling ages also indicates that our limited model resolution is insufficient to capture material heterogeneities and processes acting on smaller scales as evident in nature. A comprehensive representation of this setting certainly calls for more rigorous tuning of model parameters, but even with our generalized approach the results presented here capture important first-order properties.

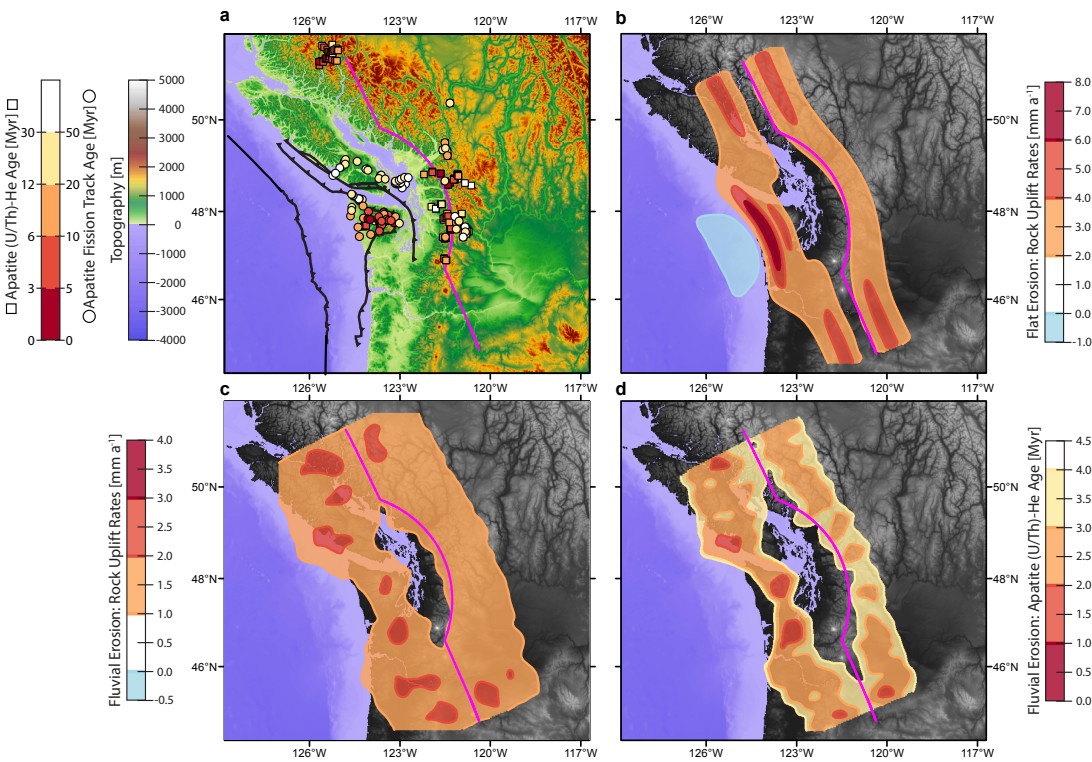

**Figure 9.** Comparison of Cascadia topography with model results: **a** Apatite (U/Th)-He (squares; Batt et al. (2001); Ehlers et al. (2006); Reiners et al. (2002, 2003); Simon-Labric et al. (2014)) and Apatite Fission track cooling ages (circles, Brandon et al. (1998); England et al. (1997); Currie and Grist (1996); Reiners et al. (2002); Johnson et al. (1986)). Major structures from Brandon et al. (1998). **b** Flat erosion (model 5) rock uplift rates. **c** Fluvial erosion (model 11) rock uplift rates. and **d** Fluvial erosion (model 11) predicted Apatite (U/Th)-He cooling ages. Model S-line in magenta for orientation. The region of increased uplift above the indenter for the flat erosion model coincides with the location of the Olympic mountains. The fluvial erosion model shows smaller variations in rock uplift rates and cooling ages that are roughly the same size as the extent of young cooling ages in the Olympics.

## 4.6 Comparison to previous studies

This work combines and builds upon previous studies. Koons et al. (2002) showed that locally enhanced erosion rates and pre-weakened crust can lead to focused exhumation. However, they used a uniform visco-plastic material for both overriding and down-going plate and a straight S-line and material influx with basal drag only on one side. This approach corresponds to our straight-slab no advance simulation. Flat erosion was applied in only a limited region, while surrounding areas were not eroded, resulting in highest rock uplift rates in this very region. In contrast, Bendick and Ehlers (2014) used an indenter-type geometry, but temperature-dependent viscous rheology for the overriding plate and uniform, flat erosion. Due to the viscous rheology, no strain localization was observed and rock uplift rates increased gradually from the edges toward the center, forming an uplift

region of several 100 km extent. While they explored effects of viscosity and indenter geometry, only full upper plate advance velocity boundary conditions were used. In contrast, the addition of frictional plasticity in this study naturally allows for strain localization. In combination with fluvial erosion, this illustrates possible mechanisms that create localized regions of uplift without relying on a priori defined structures.

## 5   Conclusions

This study investigates the thermomechanical response of upper plate deformation and erosion to subduction of a rigid indenter. We do this by exploring the effect of the presence and shape of a subducting indenter, velocity boundary conditions (i.e. amount of upper plate advance), and erosion on the resulting deformation pattern in the overriding plate. Key results from this study include:

1. For a straight subducting slab (without an indenter), shortening is accommodated by a lithospheric-scale pop-up structure composed of two broad shear zones rooting to the S-line and cutting across the entire lithospheric mantle and crust. The shear zone oriented toward the side of influx accommodates more shortening; at half advance, shear is distributed evenly.

2. Adding an indenter to the subducting plate creates another major shear system, and a strongly localized basal detachment situated above the indenter apex. Concurrently, shear across the pro shear zone is strongly reduced. The indenter detachment gives rise to a zone of localized rock uplift. Its extent along strike is governed by the indenter width. While Bendick and Ehlers (2014) showed this indenter effect solely for upper plate advance, we find that this localization effect is stronger if shortening is accommodated at least partially by lower plate subduction.

3. Under no or half upper plate advance boundary conditions, uplift above the indenter reaches a quasi-steady state with uplift rates of $7.5 \, \mathrm{mm \, a^{-1}}$ at $\sim 4 \, \mathrm{Myr}$. In case of a fully advancing upper plate, however, maximum uplift rates above the indenter are $4 \, \mathrm{mm \, a^{-1}}$ and decreases after a peak at $3.5 \, \mathrm{Myr}$, when larger amounts of shortening are accommodated by the retro-shear zone.

4. Applying a landscape evolution model for the erosional response to rock uplift modifies rock uplift through the consequent creation of topography. The increased lithostatic overburden reduces uplift rates by roughly half and distributes uplift over a wider area. Furthermore, fluvial erosion causes rock uplift to vary on a smaller (catchment-) scale than previously set by tectonics alone.

5. The deepest and fastest exhumation occurs where tectonic forcing coincides with large erosion potential. From this we conclude that both the subducting plate geometry as well as possible erosion effects need to be taken into account in order to understand the exceptional exhumation rates observed in orogen syntaxes.

*Code and data availability.*  Model output is available upon request from T. Ehlers. The software DOUAR is currently not open source. Requests for use should be made to the main author, Jean Braun (GFZ Potsdam) and Todd Ehlers (University of Tübingen).

## Appendix A: Numerical model details

Lithospheric deformation and temperatures in this study are calculated with the program DOUAR (Braun et al., 2008; Thieulot et al., 2008), a fully coupled three-dimensional thermomechanical model. Further details can also be found in Braun and Yamato (2010) and Whipp et al. (2014).

DOUAR solves the three-dimensional Stokes (creeping) flow equations for incompressible fluids, constituted by conservation of momentum (Eq. A1) and conservation of mass (Eq. A2):

$$\nabla \cdot \mu \left( \nabla \boldsymbol{V} + \nabla \boldsymbol{V}^T \right) - \nabla P = \varrho g; \tag{A1}$$

$$\nabla \boldsymbol{V} = 0, \tag{A2}$$

where $\mu$ is the material shear viscosity, $\boldsymbol{V}$ is the velocity field, $P$ the pressure, $\varrho$ is the density and $g$ gravity acceleration. The solution is computed with the finite element method using Q1P0 elements, i.e. the pressure is calculated from the velocity field by the penalty formulation (e.g. Bathe, 1982):

$$P = -\lambda \nabla \cdot \boldsymbol{V}. \tag{A3}$$

For this, conservation of mass is amended to near incompressibility with a penalty factor $\lambda$, which is typically eight orders of magnitude larger than the viscosity $\mu$. The model domain is subdivided into elements by a regular grid, on which the finite element solution is calculated.

The material properties of each element are defined by marker particles of two types, that a) track material interfaces (*surfaces*) or b) record strain and pressure (*cloud*). Particles will be created or deleted automatically to ensure both a roughly homogeneous particle density and adequate base for material property calculations (*self-adapting density*). Additionally, a third type of particles stores position, temperature and pressure for each time step. If those particles are exhumed at the surface, the p-T-t path is compiled from storage and registered at the current time step. From these paths, thermochronometric cooling ages can be calculated.

Materials can be either purely viscous or frictional visco-plastic. The viscosity $\mu$ follows a thermally activated creep law:

$$\mu = \mu_0 \dot{\varepsilon}^{1/n-1} e^{Q/nRT} \tag{A4}$$

where $\mu_0$ is the intrinsic viscosity, $\dot{\varepsilon}$ the second invariant of the deviatoric strain rate tensor, $n$ the stress exponent, $Q$ the activation energy, $R$ the gas constant and $T$ the temperature. If the stress exponent $n = 1$, the material is linear viscous and $n > 1$ denotes non-Newtonian viscosity, where viscosity increases under higher strain rates.

When plasticity is enabled, material deformation is dictated by the Mohr-Coulomb failure criterion:

$$\tau = C_0 - \sigma_n \tan \phi \tag{A5}$$

where $\tau$ is the deviatoric shear stress, $C_0$ the material cohesion, $\sigma_n$ the normal stress and $\phi$ the material angle of friction. For each model element, the effective stress $\tau_{eff}$ is calculated from strain rate:

$$\tau_{eff} = 2\mu\dot{\varepsilon}. \tag{A6}$$

If this effective stress exceeds the Mohr-Coulomb yield stress $\tau$, elemental viscosity is reduced to an effective viscosity

$$\mu_{eff} = \frac{\tau}{2\dot{\varepsilon}}, \tag{A7}$$

otherwise viscosity is kept at the initial value.

*Author contributions.* Project design and funding was done by T. Ehlers. Software development was done by M. Nettesheim with support from D. Whipp and T. Ehlers. M. Nettesheim designed and carried out simulations in close collaboration with T. Ehlers and D. Whipp. Manuscript preparation was done by M. Nettesheim and T. Ehlers with contributions from all co-authors.

*Competing interests.* The authors declare that they have no competing interests.

*Acknowledgements.* This work was supported by a European Research Council (ERC) Consolidator Grant (615703) to T. Ehlers. We also acknowledge support by Deutsche Forschungsgemeinschaft and Open Access Publishing Fund of University of Tübingen. We thank J. Braun for providing the FastScape source code used for the coupled erosion-deformation simulations, and R. Bendick for helpful discussions over the years. This manuscript benefited from thoughtful reviews by A. Replumaz and an anonymous reviewer.

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
