# Peer review of "The influence of upper plate advance and erosion on overriding plate deformation in orogen syntaxes"

_Solid Earth, 2018_

## Referee Comment (RC1) · Anonymous Referee #1 · 23 May 2018

General comments

The work by Nettesheim et al. uses numerical models to investigate the effect of subducting and overriding plate velocities on overriding plate deformation, shear zone structures and erosion patterns. The topic is of general interest, but I believe this work needs a big improvement before being suitable for publication. In terms of modelling, I think more models are needed to really understand the results and the effects of the initial setup choices. In terms of style, I think the conclusions are not clear and the message of this paper is not conveyed. Moreover, the discussions need to be expanded and re-worked, since as they are now they are mostly a repetition of the results.

[Figure]

Specific comments

I think the use of the terminology "plate corners" is used in an erroneous way. The authors show in Figure 1 what they consider plate corners. However, I do not understand the logic behind these choices. I can see how the regions highlighted by the red circles in the Aleutian and Himalayas can maybe be considered as corners, but I'm struggling to see corners in all the other circles. In South America or Indonesia, for instance, those are simply regions of curved trench, I see no corners there. And they have an opposite curvature too, which does not match the definition used at the beginning of the Introduction. Moreover, if those in South Sandwich, Mariana, Caribbean are plate corners, then why not consider corners the edges or changes in trench curvature in other subduction zones like Tonga-Kermadec, Calabria, Aegean, New Hebrides, and many more? I think the authors should be very careful in using a terminology that I found vague and wrong. Moreover, except for the places labelled in the figure, the other regions are hardly, if not never, mentioned in the manuscript, so I do not see the point of saying that those red circled regions are plate corners and then never mention them.

An other definition that I find confusing and, in my opinion, wrong is what the authors define "slab advance". First of all, it is confusing and counter intuitive that the model called "full slab advance" is the one where the slab has null velocity and the overriding plate is moving towards the trench. Secondly, Heuret and Lallemand (2005) do not define slab advance as the "migration of the overriding plate towards the down-going plate" as suggested by the authors (page 2 lines 5-6). Instead, they talk about "upper plate advance", which I believe is much more appropriate here to describe the model in which the overriding plate (and not the subducting plate) is moving towards the trench. This is not just semantic, but an important point because I am not sure that slab advancing is the cause of what the authors observe in the models. If the slab was advancing in the models forces might be distributed differently and stresses might be accommodated in different regions compared to these models in which is the upper

plate that is forced towards the trench.

Results and conclusions of this work are based on solely 3+1 models. How do the authors know the effect of the indenter geometry (paragraph 4.1) if there are no models with different indenter geometries? I think this work needs more models. The authors should at least show results from models with the 3 different types of velocities in a setup with a straight trench and no changes in slab dip. How do the shear zones forming in the overriding plate look like with a straight trench? And how do they compare to the models with a curved one? This would really help understanding the effect of the chosen geometry.

Considering the regions that this work aims to study (mostly Himalayas and Alaska), I am puzzled by the choice of the subducting plate geometry in the models. Those areas are slab edges and they have a shape that is more similar to a cusp than a smooth convex arc as modelled here. My point is that, although I do understand that models are inherently simplified and cannot be exactly like nature, I am not sure this is the best geometry to use to model those areas. The authors should discuss more in details their choice and, perhaps, even consider a different initial geometry.

The Discussion part needs to be more exhaustive. Most of the text in the Discussion is just a repetition of the results with no or little interpretation. The differences between the models are described, but why are those difference present? What is causing them? What is the effect of the chosen 3D shape? Again, a comparison with models with straight trench would help with this. For instance, would a straight trench produce slower exhumation rate? What is the effect of the chosen plate velocities? Why are these results important? How do they compare to natural cases (not only in terms of uplift rates, but also in terms of structures)? Are these 'plate corners' the only places with rapid exhumation? Can there be other causes for it other than the geometry? What is the main message of this paper? All these points are examples of what can help expanding the Discussion, improving the conclusions and the impact of the paper.

[Figure]

These are lithospheric scale models and, thus, do not include mantle flow. What do the authors think would be the effect of the mantle flow especially on the evolution of stresses and topography?

Info on velocity and temperature boundary conditions should be given for every boundary. For instance, what are the temperature conditions at sides (at x and y 0 and 800km)? What are the velocity boundary conditions at the bottom and the top? Also, from the plots it seems to me that the top boundary is free slip and not free surface. If this is the case, how is the topography computed?

I'd suggest a more comprehensive explanation on thermochronometric ages. If a reader is new to this technique will not be able to follow the paper and understand the results.

Paragraph 3.1 would be easier to follow if the 'pro-shear band', 'retro-shear band', and ' basal detachment' would be labelled explicitly in Fig. 3 and 4. Paragraph 3.2 and Figure 6. The model "full slab advance" is the only one that shows subsidence between between the structures the left and right of the S-line. This is worth mentioning and discussing.

Technical comments

Table 2 has two vsub in the header.

Fig. 1: Spell out the acronyms in the caption

P1, line 16: Change "around plate plate corners" to "around plate corners"

P2, line 18: Remove 's' from impacts: "Slab advance and erosion impacts"

P4, line 11: Specify what is the thickness range of the weak layer that decouples the plates

P5, line 13: What is the initial surface elevation?

P6, line 19: Unit of strain rate is wrong, shouldn't it be s-1?
* * *

---

## Referee Comment (RC2) · A. Replumaz (Referee) · 31 May 2018

In this paper the authors show a fully coupled thermo-mechanical numerical model to investigate the effect of a curved slab advancing on overriding plate deformation and they test different erosion scenarios on the resulting topography. This coupling between 3D thermo-mechanical models and erosion is ambitious, the subject could be very interesting as it is dealing with an emblematic problem, the curved subduction zones and the syntaxes. But as it is presented now it is a bit disappointing, as the setup is presented as global, corresponding to all the observations of figure 1, with no specific case study proposed, and I find the conclusions of the paper hard to compare with a

natural case. The authors said that their setup is similar to the Cascadia subduction zone or the Alaskan plate corner (p2, line2), in that case they have to show a map of the plates boundary and of the slab geometry in these regions, so that the reader is able to compare with the model. Having worked upon the India/Asia collision zone, for me the models presented in this paper could not be compared with the Indian slab corners in the Himalayas or with the Indonesian trench smooth curvature.

I suggest 1/ to better analyze the case studies, to properly differentiate different cases as trench curvature (Andes), slab corner (Himalayas), but also sense of curvature (convex for himalayas versus concave for andes or alps), see below analogue modeling Bajolet et al., 2013. 2/ to choose 2 very different cases to model, for example trench curvature / slab corner or concave / convex curvature. Remove the low convergence case (half slab advance name is not clear at all) which is of low interest according to me. 3/ show the erosion pattern for each case (as figure 9a), as it is an important issue of your work.

figures should be bigger, and better focus. Show the plates boundaries and find a way to represent slab geometry on figure 1. It will be better for the reader as it will be possible to see what you are talking about, and it will help to differentiate the kind of curvature/slab corner.

In this paper the authors show a fully coupled thermo-mechanical numerical model to investigate the effect of a curved slab advancing on overriding plate deformation and they test different erosion scenarios on the resulting topography.

This coupling between 3D thermo-mechanical models and erosion is ambitious, the subject could be very interesting as it is dealing with an emblematic problem, the curved subduction zones and the syntaxes. But as it is presented now it is a bit disappointing, as the setup is presented as global, corresponding to all the observations of figure 1, with no specific case study proposed, and I find the conclusions of the paper hard to compare with a natural case. The authors said that their setup is similar to the Cascadia subduction zone or the Alaskan plate corner (p2, line2), in that case they have to show a map of the plates boundary and of the slab geometry in these regions, so that the reader is able to compare with the model. Having worked upon the India/Asia collision zone, for me the models presented in this paper could not be compared with the Indian slab corners in the Himalayas or with the Indonesian trench smooth curvature.

I suggest

1/ to better analyze the case studies, to properly differentiate different cases as trench curvature (Andes), slab corner (Himalayas), but also sense of curvature (convex for himalayas versus concave for andes or alps), see below analogue modeling Bajolet et al., 2013.

2/ to choose 2 very different cases to model, for example trench curvature / slab corner or concave / convex curvature. Remove the low convergence case (half slab advance name is not clear at all) which is of low interest according to me.

3/ show the erosion pattern for each case (as figure 9a), as it is an important issue of your work.

figures should be bigger, and better focus.

Show the plates boundaries and find a way to represent slab geometry on figure 1. It will be better for the reader as it will be possible to see what you are talking about, and it will help to differentiate the kind of curvature/slab corner.

[Figure]

**Fig. 1.**

---

## Author Comment (AC1) · 10 Aug 2018

We would like to thank the reviewer for their critical assessment of our work and their constructive comments. In the following, we will address the reviewer's concerns point by point. For better readability and because of the extensive modification of this manuscript, we provide references to the differenced document rather than showing differences directly in this response letter.

Reviewer Point P 1.1 - The work by Nettesheim et al. uses numerical models to investigate the effect of subducting and overriding plate velocities on overriding plate deformation, shear zone structures and erosion patterns. The topic is of general interest,

but I believe this work needs a big improvement before being suitable for publication. In terms of modelling, I think more models are needed to really understand the results and the effects of the initial setup choices. In terms of style, I think the conclusions are not clear and the message of this paper is not conveyed. Moreover, the discussions need to be expanded and re-worked, since as they are now they are mostly a repetition of the results.

Reply: We thank the reviewer for their assessment and constructive comments to improve our work.

Reviewer Point P 1.2 - I think the use of the terminology "plate corners" is used in an erroneous way. The authors show in Figure 1 what they consider plate corners. However, I do not understand the logic behind these choices. I can see how the regions highlighted by the red circles in the Aleutian and Himalayas can maybe be considered as corners, but I'm struggling to see corners in all the other circles. In South America or Indonesia, for instance, those are simply regions of curved trench, I see no corners there. And they have an opposite curvature too, which does not match the definition used at the beginning of the Introduction. Moreover, if those in South Sandwich, Mariana, Caribbean are plate corners, then why not consider corners the edges or changes in trench curvature in other subduction zones like Tonga-Kermadec, Calabria, Aegean, New Hebrides, and many more? I think the authors should be very careful in using a terminology that I found vague and wrong. Moreover, except for the places labelled in the figure, the other regions are hardly, if not never, mentioned in the manuscript, so I do not see the point of saying that those red circled regions are plate corners and then never mention them.

Reply: We agree that the expression "plate corner" might be misleading. It is arguably not directly applicable to all regions depicted in the original Figure 1, which was a compilation of all strongly bent subducting plates that might act as rigid indenters, in order to convey the wide applicability of our presented approach. We have clarified our terminology in the introduction and replaced Figure 1 with a more detailed map of the

Cascadia subduction zone (page 3) as well as shifted the focus of our manuscript more towards this region.

Reviewer Point P 1.3 - Another definition that I find confusing and, in my opinion, wrong is what the authors define "slab advance". First of all, it is confusing and counter intuitive that the model called "full slab advance" is the one where the slab has null velocity and the overriding plate is moving towards the trench. Secondly, Heuret and Lallemand (2005) do not define slab advance as the "migration of the overriding plate towards the down-going plate" as suggested by the authors (page 2 lines 5-6). Instead, they talk about "upper plate advance", which I believe is much more appropriate here to describe the model in which the overriding plate (and not the subducting plate) is moving towards the trench. This is not just semantic, but an important point because I am not sure that slab advancing is the cause of what the authors observe in the models. If the slab was advancing in the models forces might be distributed differently and stresses might be accommodated in different regions compared to these models in which is the upper plate that is forced towards the trench.

Reply: We acknowledge the reviewer's concern about the term "slab advance", which is a question to the frame of reference. In our chosen, moving frame of reference centered on the indenter's toe, trenchward motion of the overriding plate is equal to a slab or trench advance in a frame of reference fixed to the overriding plate. Physically, forces, strain rates and uplift velocities are not affected by this uniform addition of horizontal velocity (i.e. a Galileian transformation). Nevertheless, in order to avoid confusing the reader, we have replaced the term "slab advance" with "upper plate advance" throughout the manuscript. See the reworded definition in the introduction on page 3, line 1.

Reviewer Point P 1.4 - Results and conclusions of this work are based on solely 3+1 models. How do the authors know the effect of the indenter geometry (paragraph 4.1) if there are no models with different indenter geometries? I think this work needs more models. The authors should at least show results from models with the 3 different types

of velocities in a setup with a straight trench and no changes in slab dip. How do the shear zones forming in the overriding plate look like with a straight trench? And how do they compare to the models with a curved one? This would really help understanding the effect of the chosen geometry.

Reply: We understand this to be one of the central points in the reviewer's concerns and agree that additional models will help the reader understand the effect of an indenter-type geometry more clearly. We have added three models with a straight trench for our three velocity boundary conditions. Results are shown in Figures 3 (page 14) and 4 (page 15) in section 3.1.1 (pages 9 ff.), as well as in supplemental figure S1. Moreover, we have added two more models with a narrower and a wider indenter, shown in figure 8 in section 3.2 (pages 19 and 17, respectively). These results are discussed in section 4.3 (pages 21 f.).

Reviewer Point P 1.5 - Considering the regions that this work aims to study (mostly Himalayas and Alaska), I am puzzled by the choice of the subducting plate geometry in the models. Those areas are slab edges and they have a shape that is more similar to a cusp than a smooth convex arc as modelled here. My point is that, although I do understand that models are inherently simplified and cannot be exactly like nature, I am not sure this is the best geometry to use to model those areas. The authors should discuss more in details their choice and, perhaps, even consider a different initial geometry.

Reply: As mentioned above, we have replaced figure 1 with a detailed map showing the subducting slab geometry of the Cascadia subduction zone in order to document our choice of subduction plate geometry (page 3). Moreover, we have run and discussed additional models with wider and narrower indenter geometry (section 3.2 on page 17 and section 4.3 pages 21 f.). Nevertheless, we do not aim to reproduce any particular region, but rather study the effect of a rigid indenter on upper plate deformation. In this, our focus admittedly lies more on regions that are well studied and show exceptional properties, and thus interest a broader audience. Our findings are transferable to cusplike, asymmetric subduction zones. Figure R1 shows that the main difference between a cusp-type geometry, modeled after the Alaskan plate corner, and symmetric indenter geometry lies in the transition to a transfer fault on one side. These results agree with observed fault patterns (see Koons et al. (2010) and references therein). However, we think that adding this specific plate geometry will go beyond our generic approach and intended scope of this manuscript.

Reviewer Point P 1.6 - The Discussion part needs to be more exhaustive. Most of the text in the Discussion is just a repetition of the results with no or little interpretation. The differences between the models are described, but why are those difference present? What is causing them? What is the effect of the chosen 3D shape? Again, a comparison with models with straight trench would help with this. For instance, would a straight trench produce slower exhumation rate? What is the effect of the chosen plate velocities? Why are these results important? How do they compare to natural cases (not only in terms of uplift rates, but also in terms of structures)? [...]

Reply: We thank the reviewer for his helpful suggestions to improve the discussion section. We have added straight-slab models (Figs. 3 and 4 on pages 14 and 15, respectively) as well as models with different indenter geometries (Fig. 8 on page 19) to this manuscript in order to discuss the questions raised here in more depth. Furthermore, we have added a model with twice as fast convergence (S2), which shows that in our setup and to our line of investigation, the total rate of convergence is less important than its partitioning in subduction and upper plate advance. Alongside the addition of these models, we have substantially revised the discussion section (see sections 4.3 and 4.4 on pages 21ff.) . In doing that, we have put more emphasis on the underlying mechanisms that cause the observed results. Moreover, we relate our results more closely to natural observations, including the newly added figure 10 on page 25. For example, detachment- like shallow faults dipping toward the subducting plate, similar to our models, have been observed in South America. Likewise, the shallow detachment dipping towards the continent, joined by a steeper fault at depth, in our full advance

models, well matches the geometry of the Hurricane Ridge Fault of the Olympic mountains as described by Willett (1999). Due to our simplified approach, such comparisons can only be done qualitatively, but we're confident these additions will help the reader transfer our results to nature more easily.

Reviewer Point P 1.7 - [...] Are these "plate corners" the only places with rapid exhumation? Can there be other causes for it other than the geometry? What is the main message of this paper? [...]

Reply: The discussed regions are certainly not the only ones with rapid exhumation. For example, exhumation in Taiwan has been found to range from 3 to 5 mm a 1 over the last 2-3 Ma (e.g. Fuller et al., 2006; Willett et al., 2003). However, the aim of this study is not to investigate rapid erosion in general, but to further investigate the effects of an indenter geometry and its contribution to localized uplift as observed in orogen syntaxes. In previous studies, two main hypotheses were models, the effect of an indenter geometry signifies an important contribution to localized uplift, as proposed by Bendick and Ehlers (2014), and cannot be neglected when investigating regions with a curved subducting plate. We have edited the discussion and conclusion accordingly, to make this point clearer to the reader.

Reviewer Point P 1.8 - [...] All these points are examples of what can help expanding the Discussion, improving the conclusions and the impact of the paper. These are lithospheric scale models and, thus, do not include mantle flow. What do the authors think would be the effect of the mantle flow especially on the evolution of stresses and topography?

Reply: We agree that mantle flow may have a strong effect. Mantle flow and slab penetration determine relative plate motion (e.g. Heuret and Lallemand, 2005; Faccenna et al., 2013). This causes partitioning of shortening into subduction and upper plate motion, which is one of main parameters in this study. We added a related note to the methods section on page 3, line 1. We also understand that mantle vertical movements

that are not included might be important for geodynamics. However, we assume their role limited in the subduction convergent settings comparing to extensional ones.

Reviewer Point P 1.9 - Info on velocity and temperature boundary conditions should be given for every boundary. For instance, what are the temperature conditions at sides (at x and y 0 and 800km)? What are the velocity boundary conditions at the bottom and the top? Also, from the plots it seems to me that the top boundary is free slip and not free surface. If this is the case, how is the topography computed?

Reply: We have added the desired information and furthermore revised methods sections 2.2.4 and 2.2.5 to convey information more clearly (see pages 7 f. and also Fig. 2 on page 5).

Reviewer Point P 1.10 - I'd suggest a more comprehensive explanation on thermochronometric ages. If a reader is new to this technique will not be able to follow the paper and understand the results.

Reply: We have added an explanatory paragraph, see section 2.2.6 on page 8

Reviewer Point P 1.11 - Paragraph 3.1 would be easier to follow if the "pro-shear band", "retro-shear band", and "basal detachment" would be labeled explicitly in Fig. 3 and 4. Paragraph 3.2 and Figure 6.

Reply: In order to clarify this for the reader, we have adopted a consistent terminology of "pro shear zone", "retro shear zone" and "indenter detachment" throughout the manuscript and added labels of p , r , and i , respectively, to all figures. See e.g. Figure 4 on page 15.

Reviewer Point P 1.12 - The model "full slab advance" is the only one that shows subsidence between the structures the left and right of the S-line. This is worth mentioning and discussing.

Reply: The subsidence developing in the full advance case is caused by isostatic relaxation. In the early stages of the model, upward material flow along the subduction interface causes crustal uplift and thinning by flat erosion. When the retro-shear zone becomes more localized and accommodates more deformation, the upward push ceases and isostatic relaxation causes subsidence. This feature is reminiscent of the flexural subsidence of the Chilean forearc by thrust or reverse faulting in the late Cenozoic (Horton, 2018a, b, and references therein). See page 22, line 27 f.

Technical Comments

Reviewer Point P 1.13 - Table 2 has two vsub in the header. Reply: Fixed, see table 1 on page 9.

Reviewer Point P 1.14 - Spell out the acronyms in the caption Reply: Figure 1 has been replaced

Reviewer Point P 1.15 - Change "around plate plate corners" to "around plate corners" Reply: Done, see page 1, line 18.

Reviewer Point P 1.16 - Remove 's' from impacts: "Slab advance and erosion impacts" Reply: Done, see page 3, line 11.

Reviewer Point P 1.17 - Specify what is the thickness range of the weak layer that decouples the plates Reply: Values added, see page 6, line 10.

Reviewer Point P 1.18 - Unit of strain rate is wrong, shouldn't it be s-1? Reply: Fixed, see page 9, line 12.

Please also note the supplement to this comment:
https://www.solid-earth-discuss.net/se-2018-14/se-2018-14-AC1-supplement.pdf

———————————————————

[Figure]

Observations
(Slab 1.0)

model geometry

**Depth [km]**

**Topography [m]**

-1000
-2000
-3000
-4000

km
0 75 150  300  450  600

US  CA

X [km]
0    200   400   600   800

800

600

X [km]
0    200   400   600   800

800

600

on Depth [km]
20.0

16.0

12.0

[mm a⁻¹]
8.0

6.0

4.0

**Supplement:**

[revised manuscript text omitted]

**Figure S1.** Comparison of velocities and temperatures between straight-slab (left column) and intenter-type (right column) flat erosion models (models 1–6). Colored lines denote material layer boundaries. Panels **a1**–**b3**: Background colors denote vertical velocity component, material flow lines are colored by total velocity. Panels **a1**–**b3**: Background color show temperatures. Colored lines denote material layer boundaries. Plots show the increased and focused uplift around $x = 250\,\mathrm{km}$ created by the indenter and the corresponding temperature anomalies, which mostly correspond to deformation.

[Figure]

**Figure S2.** Comparison of standard and twice as fast convergence models (models 5 and 7) after 4 and 2 Myr modeling time, respectively (equal amount of material added to domain). Panels **a** and **c** show rock uplift rates, panels **d** and **e** strain rates at $y =400$ km and panel **b** shows surface rock uplift rates along $y =400$ km slice. Please note that all scales for fast convergence model are doubled (gray labels), but relative distribution of rock uplift and strain rates is almost the same.

[Figure]

**Figure S3.** Comparison of rock uplift and exhumation depths with fluvial versus total erosion for no (model 10, upper half) and full (model 12, lower half) upper plate advance. **a** and **h** surface elevations, **b** and **i** rock uplift rates, and **c** and **j** exhumation depth after 6.0 Myr modeling time. Uplift is focused more toward the center for the no advance and more to the front and back for the full advance fluvial erosion model. **d**, **e**, **f**, and **g** show the respective flat erosion results (models 4 and 6) for rock uplift rates and exhumation depth for comparison. Note the range is increased by factor 2 for those four plots (gray labels).

---

## Author Comment (AC2) · 10 Aug 2018

We would like to thank the A. Replumaz for her critical review of our work and her constructive comments. In the following, we will address her concerns point by point. For better readability and because of the extensive modification of this manuscript, we provide references to the differenced document (see response to reviewer 1) rather than showing differences directly in this response letter.

Reviewer Point P 2.1 - In this paper the authors show a fully coupled thermo-mechanical numerical model to investigate the effect of a curved slab advancing on overriding plate deformation and they test different erosion scenarios on the resulting topography. This coupling between 3D thermo-mechanical models and erosion is ambitious, the subject could be very inter- esting as it is dealing with an emblematic problem, the curved subduction zones and the syntaxes. But as it is presented now it is a bit disappointing, as the setup is presented as global, corresponding to all the ob- servations of figure 1, with no specific case study proposed, and I find the conclusions of the paper hard to compare with a natural case. The authors said that their setup is similar to the Cascadia subduction zone or the Alaskan plate corner (p2, line2), in that case they have to show a map of the plates boundary and of the slab geometry in these regions, so that the reader is able to compare with the model. Having worked upon the India/Asia collision zone, for me the models presented in this paper could not be compared with the Indian slab corners in the Himalayas or with the Indonesian trench smooth curvature.

Reply: We thank A. Replumaz for her assessment of and constructive suggestions to improve our work. We understand her main concern lies in the limited transfer to natural settings. We have edited our discussion section to include more references to natural systems and added Figure 10 on page 25 to illustrate the relation of our results to nature. Figure 1(page 3) now shows the geometry of Cascadia subduction zone. As discussed above (see figure R1), we are convinced that our results are also transfer- able to asymmetric cusp-like geometries. We agree that detailed adjustment of initial and boundary conditions is required before conclusions about a specific subduction setting can be drawn.

Reviewer Point P 2.2 - I suggest 1/ to better analyze the case studies, to properly differentiate different cases as trench curvature (Andes), slab corner (Himalayas), but also sense of curvature (convex for himalayas versus concave for andes or alps), see below analogue modeling Bajolet et al., 2013. I suggest [...] 2/ to choose 2 very different cases to model, for example trench curvature / slab corner or concave / convex curvature.

Reply: This point is closely linked to concerns expressed by the first reviewer. We

intentionally limit ourselves to convex plate corners, where flexural stiffening of the downgoing plate (see Mahadevan et al., 2010) is thought to occur, in order to specifically investigate a possible focusing effect on upper plate deformation as proposed by Bendick and Ehlers (2014). We agree with both reviewers that additional models will be very helpful to illustrate this effect. We have added three models with a straight trench well as two more models with a narrower and a wider indenter. Results are shown in figures 3 (page 14), 4 (page 15), and 8 (page 19) in sections 3.1 and 3.2 on pages 9 ff. and 17. See also discussion in sections 4.3 on page 21 f.

Reviewer Point P 2.3 - Remove the low convergence case (half slab advance name is not clear at all) which is of low interest according to me.

Reply: Perhaps we were unclear in the manuscript as it seems the half advance case may have been misunderstood. For example, the half-advance scenario is quite similar to the convergence between India and Eurasia in the Himalaya, where India moves into Eurasia at around 45-50 mm/a, but only about 40-50% of that convergence is shortening in the Himalaya (e.g. Ader et al., 2012). In addition, we note that the convergence rate is the same in all models presented in this study. To ensure readers are not misled, we have edited the description of our boundary conditions (see section 2.2.4 on page 2.2.4) and expanded table 1 (page 9) in the revised version in order to provide a clearer explanation of the different boundary conditions and their geodynamic implication.

Reviewer Point P 2.4 - I suggest [...] 3/ show the erosion pattern for each case (as figure 9a), as it is an important issue of your work.

Reply: We have added the requested models to our study, results are shown in figure S3.

Reviewer Point P 2.5 - figures should be bigger, and better focus. Show the plates boundaries and find a way to represent slab geometry on figure 1. It will be better for the reader as it will be possible to see what you are talking about, and it will help to

differentiate the kind of curvature/slab corner.

Reply: Figures 1 (page 3), 3 (page 14) and 4 (page 15) have been replaced in the revised version. Additionally, we have added four figures to the supplement for a more detailed understanding of the reader. We will request large figure sizes in the final publication for best perception.

––––––––––––––––––––––––––––––

---

## Editor Comment (EC1) · M.B. Allen (Editor) · 13 Aug 2018

Great - now please hit all buttons on the system to make it clear it's the final submission. The SE editorial process is like the EU: great intentions, but a bit of a swamp to wade through. And the fact that this brief note merits the status of an official online document with its own doi is just bizarre (https://doi.org/10.5194/se-2018-14-EC1 for the record).

regards Mark
* * *

---

## Author Response (AR2)

**Response to anonymous reviewer 2**
* * *
**Reviewer 2**

**Reviewer Point P 2.1** — I found that the paper has improved a lot, the text is clearer, the goal and the target are clearer. I have only one major comment and 2 minor recommendations on the figures:

**Reply**: We thank the reviewer for their renewed assessment of our work.

**Reviewer Point P 2.2** — why so many outputs of your model on thermochronology, and no comparison with the data ? remove most of the figures for different thermochronometers, or put the data on figure 10 and compare with the models !

**Reply**: We agree with the reviewer that showing observed data will help the reader not familiar with the Olympic Mountains to better assess our work. Thus, we have added Apatite He and Apatite FT cooling ages to Figure 9 and adjusted the correspoding discussion paragraph on page 19. Additionally, we have moved the previous Figure 6 to the supplement, since it is not as important to the main message of this study.

**Reviewer Point P 2.3** — figure 3 is still very complex, too many details. For this presentation of the results you should only show "b" and "c" full page

**Reply**: Following the reviewer's suggestions, we have changed this Figure to only show the indenter-geometry model (see page 10) and modified the text in section 3.1.1 on pages 7ff. accordingly.

**Reviewer Point P 2.4** — idem for figure 5, show only the stage "8Ma", try to make the cross-section bigger (maybe in column ?), for me it is really a key result, the different geometry of the faults for the different setting, and it is not well presented

**Reply**: In order to better highlight this comparison of different scenarios, we have replaced the multiple maps of rock uplift rates with a set of line plots, see the updated Figure 5 on page 12.

[revised manuscript text omitted]